# Pathophysiological Role and Medicinal Chemistry of A_2A_ Adenosine Receptor Antagonists in Alzheimer’s Disease

**DOI:** 10.3390/molecules27092680

**Published:** 2022-04-21

**Authors:** Stefania Merighi, Pier Andrea Borea, Katia Varani, Fabrizio Vincenzi, Alessia Travagli, Manuela Nigro, Silvia Pasquini, R. Rama Suresh, Sung Won Kim, Nora D. Volkow, Kenneth A. Jacobson, Stefania Gessi

**Affiliations:** 1Department of Translational Medicine and for Romagna, University of Ferrara, 44121 Ferrara, Italy; vrk@unife.it (K.V.); fabrizio.vincenzi@unife.it (F.V.); alessia.travagli@edu.unife.it (A.T.); manuela.nigro@unife.it (M.N.); 2Department of Chemical, Pharmaceutical and Agricultural Sciences, University of Ferrara, 44121 Ferrara, Italy; bpa@unife.it (P.A.B.); silvia.pasquini@unife.it (S.P.); 3Molecular Recognition Section, Laboratory of Bioorganic Chemistry, National Institute of Diabetes and Digestive and Kidney Diseases, Bethesda, MD 20892, USA; rama.ravi@nih.gov; 4Laboratory of Neuroimaging, National Institute on Alcohol Abuse and Alcoholism, National Institutes of Health, Bethesda, MD 20892, USA; sunny.kim@nih.gov (S.W.K.); nvolkow@nida.nih.gov (N.D.V.); 5National Institute on Drug Abuse, National Institutes of Health, Bethesda, MD 20892, USA

**Keywords:** A_2A_ receptors, A_2A_ antagonist discovery, Alzheimer’s disease, memory loss, neurodegeneration

## Abstract

The A_2A_ adenosine receptor is a protein belonging to a family of four GPCR adenosine receptors. It is involved in the regulation of several pathophysiological conditions in both the central nervous system and periphery. In the brain, its localization at pre- and postsynaptic level in striatum, cortex, hippocampus and its effects on glutamate release, microglia and astrocyte activation account for a crucial role in neurodegenerative diseases, including Alzheimer’s disease (AD). This ailment is considered the main form of dementia and is expected to exponentially increase in coming years. The pathological tracts of AD include amyloid peptide-β extracellular accumulation and tau hyperphosphorylation, causing neuronal cell death, cognitive deficit, and memory loss. Interestingly, in vitro and in vivo studies have demonstrated that A_2A_ adenosine receptor antagonists may counteract each of these clinical signs, representing an important new strategy to fight a disease for which unfortunately only symptomatic drugs are available. This review offers a brief overview of the biological effects mediated by A_2A_ adenosine receptors in AD animal and human studies and reports the state of the art of A_2A_ adenosine receptor antagonists currently in clinical trials. As an original approach, it focuses on the crucial role of pharmacokinetics and ability to pass the blood–brain barrier in the discovery of new agents for treating CNS disorders. Considering that A_2A_ receptor antagonist istradefylline is already commercially available for Parkinson’s disease treatment, if the proof of concept of these ligands in AD is confirmed and reinforced, it will be easier to offer a new hope for AD patients.

## 1. Introduction

Adenosine is a ubiquitous nucleoside derived largely from the hydrolysis of ATP, which acts as an extracellular autacoid to regulate a wide number of pathophysiological functions. The intracellular production of adenosine arises from the action of cytosolic 5′-nucleotidase on AMP or the action of SAH hydrolase on S-adenosyl-homocysteine (SAH) hydrolysis. Conversely, an extracellular pathway predominates when cells undergo increased metabolic activity or stress, e.g., in pathological conditions. In this case, the main source of adenosine is from the activity of specific enzymes known as ectonucleoside triphosphate diphosphohydrolases (CD39s), which convert ATP into AMP, and ecto-5′-nucleotidase (CD73), which dephosphorylates AMP to produce adenosine [1]. Upon generation, the extracellular nucleoside enters cells through both concentrative and equilibrative nucleoside transporters, CNTs (SLC28 family) and ENTs (SLC29 family), respectively. These proteins are crucial to fine-tuning adenosine levels, with the ENTs playing the more important role. CNT proteins increase adenosine from the extracellular to intracellular pools driven by the sodium gradient, while ENTs allow bidirectional transport, following the nucleoside concentration gradient across the membrane. Normally, adenosine produced extracellularly enters into cells by these transporters, while during hypoxia this flux can be reversed [2].

The amount of extracellular adenosine is also regulated by its metabolism through the enzyme adenosine deaminase (ADA), which transforms it into inosine. This type of regulation occurs also in the intracellular pool, where adenosine may be either converted into inosine by intracellular ADA or phosphorylated to AMP by adenosine kinase (ADK). Due to the different K_m_ of these enzymes for adenosine, with ADK being more potent than ADA, under physiological conditions AMP generation is predominant, while in stressful environments its conversion to inosine principally occurs [3].

Adenosine effects are due to the stimulation of four transmembrane receptors, termed A_1_, A_2A_, A_2B_ and A_3_ adenosine receptors (ARs) [4]. These AR subtypes belong to the family of G protein-coupled receptors, either G_i_-(A_1_, A_3_) or G_s_-(A_2A_, A_2B_) coupled. Altogether, they are balanced in modulating adenylyl cyclase activity, with inhibitory or stimulatory effects depending on G_i_ or G_s_ coupling, respectively. In addition, the A_2A_ subtype in the striatum triggers recruitment of G_olf_ proteins [5,6]. Furthermore, the ARs modulate mitogen-activated protein kinases (MAPK) p38, ERK1/2, and JNK1/2 as well as Akt phosphorylation, thus regulating important intracellular signal transduction pathways, potentially affecting a wide number of pathophysiological functions [7,8,9,10,11]. 

Although all ARs are potential pharmaceutical targets for a variety of human diseases, the A_2A_ subtype is the one for which the understanding of biological relationships and the discovery of potent and selective ligands are further along a translational path. The first clinically approved selective A_2A_AR antagonist istradefylline, a caffeine analogue, is now on the market for Parkinson’s disease (PD) [12,13]. Therefore, this review will focus on a survey of recent literature on the A_2A_ adenosine receptor as a druggable target for another important neurodegenerative disorder, i.e., Alzheimer’s disease (AD), considering the present technological state and directions for improving drug discovery and development from both biological and chemical perspectives. 

## 2. A_2A_ Receptors in AD

AD is a multifactorial neurodegenerative pathology that is experiencing exponential growth in the aging population, which is expected to triple by 2050 [14]. Its pathological indications include a progressive appearance of amnesic disabilities, associated with aphasia, apraxia, and agnosia. Within the scope of this disease are behavioral problems, consisting of problem solving and orientation disorders. Current pharmacological strategies to treat AD patients are to increase synaptic acetylcholine (ACh) concentrations, using the cholinesterase inhibitors donepezil, rivastigmine, galantamine, or to reduce glutamate excitotoxicity through memantine [15,16]. Unfortunately, these drugs offer merely symptomatic relief, and they do not impede disease progression, thus making the search for new therapeutic strategies urgent. Various novel pharmacological targets aim to accelerate the clearance of β amyloid (Aβ) peptide and tau protein, or to impede neurodegenerative molecular pathways [17,18]. This is an ambitious challenge, but a new drug targeting Aβ protein accumulation, named aducanumab, recently entered the market, with still controversial clinical efficacy [18,19,20]. However, due to the failure of numerous clinical trials with drugs directed towards Aβ protein, it is questionable whether β-amyloid is certainly an effective target. Another strategy for the treatment of AD concerns tau neurofibrillary tangles that are strongly correlated with local neurodegeneration and cognitive impairment in AD. As a result, several approaches for lowering tau pathogenicity have been suggested, and tau-targeted active and passive immunotherapies are now being investigated for safety and effectiveness in patients with early-stage AD. However, because no clinical trials have been finished to date, no assumptions about the therapeutic effects of therapies for tau pathology can be established [16]. Interestingly, caffeine, the most commonly ingested A_2A_ receptor antagonist, as well as genetic elimination of A_2A_ receptors in mice, decreased hippocampus tau hyperphosphorylation, fought neuroinflammation, and restored the associated memory loss [13].

Indeed, the A_2A_ adenosine receptor may represent a novel and pleiotropic protein through which it is possible to alter the disease course. The A_2A_ adenosine receptor has been extensively characterized biochemically and structurally, with many X-ray and cryo-EM structures now available [12]. A lengthy intracellular carboxy-terminal tail, where phosphorylation and palmitoylation activities may cause receptor desensitization and internalization, is a structural feature of the A_2A_ subtype that other family members lack [21]. This subtype forms heteromers with other receptors, such as A_1_ adenosine and D_2_ dopamine receptors, that have different pharmacological properties compared to the receptor monomers [22]. In the central nervous system, the A_2A_ receptor is found at the highest levels in striatum, olfactory tubercle, and immune system, while lower amounts are present in the cerebral cortex, hippocampus and vasculature [23,24]. In any case, low expression does not imply lesser importance; indeed, cerebrocortical A_2A_ receptors account for most of the more relevant effects attributed to brain A_2A_ receptors (e.g., mood, memory). Likewise, proper recognition must be given to the sparse but highly significant cortical A_2A_ receptors.

A_2A_ receptors in the brain are mostly detected in glutamatergic synapses, where the major consequence of their activation at the presynaptic level is reflected in increased glutamate release, which contributes to excitotoxicity. Indeed, under physiological conditions, hippocampal glutamate release is under the control of an inhibitory A_1_ receptor that, following adenosine increase, is counteracted by a presynaptic A_2A_ receptor resulting in the release of this excitatory neurotransmitter, possibly through induction of PKA-dependent calcium currents [25,26]. Postsynaptically, the A_2A_ receptor promotes long-term potentiation (LTP) by facilitating AMPA receptor-mediated currents, NMDA phosphorylation that is mGluR5-dependent and a Ca^2+^ increase [27,28,29,30]. The A_2A_ receptor reduces glutamate uptake in astrocytes, through modulating the expression of glutamate transporters, GLT-1 and GLAST, and induces astrocytic proliferation and activation. In microglia it plays an important role in the induction of proliferation, activation, process retraction and the production of molecules involved in inflammation, such as NO, PGE2, IL1β and cyclooxygenase [31] (Figure 1). In addition to affecting the inflammatory response, A_2A_ receptor promotes BDNF and GDNF secretion from microglia and astrocytes, respectively, in response to LPS [1].

It is widely recognized that glutamatergic synaptic failure and temporal lobe degeneration are the earliest signs of cognitive decline, preceding the production of Aβ plaques and tau-derived neurofibrillary tangles [32]. Indeed, individuals with mild cognitive impairment (MCI) and early AD show synaptic loss in the hippocampus and posterior cingulate gyrus, indicating that this event is the starting point for memory loss [33,34,35]. Therefore, A_2A_ adenosine receptor engagement in glutamatergic synaptic physiology has suggested a relationship between this subtype and AD. The A_2A_ adenosine receptor expression in both hippocampal neurons and astrocytes is elevated in aging, AD animal models, and AD patients, elevating glutamate release, calcium influx, the LTP-to-long-term depression transition, and cognitive decline [26,36,37,38,39,40,41,42,43,44,45,46]. Overall, this evidence indicates that the important contribution of the A_2A_ adenosine receptor to synaptic and neuronal damage by increasing glutamate excitotoxicity and neuroinflammation, as evidenced by the neuroprotection provided by genetic or pharmacological A_2A_ adenosine receptor blockade. Indeed, A_2A_ receptor antagonism reduces hippocampus-dependent memory impairment and LTP changes in aged animals and AD models [36,47,48,49,50,51]. In addition, A_2A_ adenosine receptor genetic knockdown can ameliorate synaptic damage present in AD models, and numerous studies have found that A_2A_ adenosine receptor antagonists are effective at improving cognition after synaptic loss in AD animal models, providing a way to combat synaptic toxicity [40,52,53,54]. Adenosine levels in postmortem AD brains are higher in the parietal and temporal lobes than in the frontal cortex, implying enhanced A_2A_ receptor activation in overexpressing areas [55]. A study of A_2A_ receptor photoaffinity labeling in post-mortem striatal membranes provided the first hint that the A_2A_ receptor could be enhanced in the AD brain [56]. Indeed, in comparison to healthy, post-mortem subjects, brains from AD patients had a higher density of A_2A_ receptors. More interestingly, in comparison to both frontal grey matter and frontal white matter regions in AD but not in normal participants, the overexpression was observed primarily in the hippocampus and entorhinal areas, suggesting that the upregulation is likely connected to the existence of AD pathology. Indeed, A_2A_ expression appears to replicate the same AD scenario that extends in the brain from hippocampal/entorhinal structures to other cortical areas, with the hippocampal/entorhinal cortex having the highest density of pathology. As a result, the frontal white matter region, where the AD pathology is less prominent, has the lowest A_2A_ expression [24]. This leads to the idea that, once reliable PET ligands are validated, assessing A_2A_ receptor density in diseased brain circuits may become an essential biomarker of vulnerability to detect development of AD [57].

A further crucial component of A_2A_ adenosine receptor action in AD is its control of neuroinflammation via glial cells. The A_2A_ adenosine receptor was shown to be overexpressed in the astrocytes of AD patients, and its genetic silencing in both young and old mice improved long-term memory [40,58,59]. A_2A_ receptors in astrocytes were required for fine-tuning inhibitory and excitatory regulation of synaptic transmission through GABA and glutamate uptake. A_1_ and A_2A_ receptors, which can occur together as heterodimers, controlled GABA transport in different ways, with the A_1_ blocking and the A_2A_ boosting it [60]. Furthermore, glutamate uptake was hampered by A_2A_ receptor activation, which is required for the decrease of synaptic glutamate transporters GLAST and GLT-I that is produced by Aβ peptide [61]. An A_2A_ receptor antagonist or receptor knockout prevented the Aβ-induced decrease in glutamate transporters and astrogliosis. A_2A_ receptor overexpression causes significant transcriptional alterations, primarily impacting genes involved in immune responses, angiogenesis, and cell activation. Treatment with SCH58261 (Figure 2), a selective A_2A_ antagonist, restored the expression levels of numerous inflammatory and astrocytic activation-related genes. This reinforces the concept that blocking the A_2A_ receptor might reverse some astrocytic disorders caused by aberrant A_2A_ receptor expression, implying that receptor antagonists could be useful in A_2A_ receptor-induced transcriptional dysregulation, inflammation, and astrogliosis. Altogether, these findings shed light on the possible influence of A_2A_ receptor upregulation on astrocyte transcriptional dysregulation, paving the way for the development of A_2A_ receptor antagonists as a potential therapy for AD [62]. Activated microglia, in addition to astrocytes, are relevant cells for neuroinflammation and, interestingly, they overexpress the A_2A_ receptor. Microglia in AD also express the NMDA receptor, which is a crucial target in the fight against AD and might be counteracted by inhibiting the A_2A_ receptor through their crosstalk in heteromeric structures [63,64]. Furthermore, by inhibiting A_2A_ receptor-mediated cytokine production, this strategy may be able to reduce memory loss by attenuating neuroinflammation [65,66]. 

We already have proof that blocking A_2A_ improves memory in humans. As a matter of fact, caffeine, the most commonly ingested A_2A_ receptor antagonist, can ameliorate cognition providing the evidence for the A_2A_ receptor’s relevance in AD through different strategies. Indeed, caffeine decreased hippocampal tau hyperphosphorylation, attenuated neuroinflammation and reversed the associated memory loss [12,50,53,82]. The Effect of CAFfeine on Cognition in Alzheimer’s Disease (CAFCA) is under investigation in the Phase 3 clinical trial NCT04570085. It is a multicenter, randomized, double-blind, placebo-controlled trial evaluating the effect of a 30-week caffeine treatment on cognition in AD at beginning to moderate stages (MMSE 16–24). The results of human research will help to clarify the function of caffeine in the prevention of AD. In addition, istradefylline, a potent and selective A_2A_ antagonist, is already in use in the clinic for another neurodegenerative condition, PD. It was approved earlier in Japan and Korea, and in August 2019 in the United States after passing various clinical safety and efficacy tests (as Nouriast) [73,83,84,85]. As for its role in AD, low istradefylline doses improved spatial memory and habituation in animal models of AD, providing important confirmation that A_2A_ receptor blockade could be a novel target for countering cognitive deficits in AD patients [59,71]. This finding shows that A_2A_ receptor-targeting drugs, such as istradefylline and hopefully novel ones, have a great deal of promise for treating dementia.

## 3. Chemistry of A_2A_ Receptor Antagonists in Clinical Trials

Many selective A_2A_ antagonists have been reported and some have entered clinical trials (Table 1) [12,67,68,86] for either neurodegenerative diseases or cancer immunotherapy (including by co-administration with checkpoint inhibitors and other cancer immunotherapies). It is to be noted that A_2B_ antagonists and mixed A_2A_/A_2B_ antagonists are also being evaluated in cancer immunotherapy [80]. Although not covered in this review, A_2B_ antagonists may also have neuroprotective properties [87]. The naturally occurring alkylxanthine caffeine, a weak non-selective adenosine antagonist, is in a Phase 3 clinical trial for AD in France [50]. Recently, the analogue of caffeine with full deuteration of the three methyl groups was shown to have a prolonged in vivo plasma half-life in the rat (from 1.9 h to 5 h) without losing adenosine receptor binding affinity [88]. 

Istradefylline **3** [72], a moderately potent (pKi 7.44) and selective A_2A_ antagonist that is structurally related to caffeine (Figure 2), is the only such ligand that is FDA-approved for clinical use, i.e., as a co-therapy with L-DOPA in PD treatment. A total of 23 clinical trials have been carried out to probe the safety and efficacy of istradefylline in PD and also to study the effects of hepatic impairment and antibiotic rifampicin on its metabolism.

Unlike some of the more recently introduced A_2A_ antagonists for cancer, tozadenant **6** (SYN115) was initially developed for treatment of PD and later for cocaine addiction [67,89,90,91]. Tozadenant has been studied both as a PD monotherapy and as a cotherapy with L-DOPA or dopamine agonists. Clinical efficacy in reducing off-time was observed without severe adverse effects in initial clinical trials, but a Phase 3 clinical trial turned disastrous with seven cases of sepsis out of 890 patients and six fatalities [86]. Therefore, its clinical development was terminated in 2018. Its potential use in treating cocaine addiction was previously studied [89], but there was insufficient clinical efficacy to proceed with this target. Tozadenant analogues were recently reported as A_2A_ antagonists, including ^18^F-labeled radiotracers for potential PET imaging [92]. Vipadenant **7** [67] is a potent A_2A_ antagonist (pKi 8.89) that was introduced for PD [74,93] and its development halted due to safety issues [68], but it later transitioned to use in cancer immunotherapy. Other ^18^F-labeled A_2A_ antagonists related to vipadenant were reported by Lai et al. [94]. Preladenant **12** followed a similar developmental path, first being introduced for PD and later repurposed to cancer [68,95]. The ^11^C forms of a predecessor of preladenant, i.e., SCH442416 **10**, and preladenant itself, have been shown to be useful as A_2A_ receptor PET imaging agents [94]. The use of structure-based approaches has accelerated the discovery of A_2A_ antagonists. The earliest success of a structurally designed A_2A_ antagonist was imaradenant **22**. It was initially discovered using an inactive conformation-stabilized A_2A_ receptor, enabled by the ability to readily determine the three-dimensional structure of A_2A_ antagonist-bound receptors by strategic stabilizing mutations [96]. Early in its development it was positioned for use in treating attention deficit hyperactive disorder (ADHD), but it was repurposed for cancer immunotherapy. Taminadenant **24** and ciforadenant **25** are being developed for cancer immunotherapy [77,78], but ciforadenant has a long history of clinical development, initially for CNS application. An analogue of taminadenat (PBF-999, structure not shown) displayed dual A_2A_ receptor and phosphodiesterase (PDE)10 activity and is now in a clinical trial for cancer [68,94].

The role of pharmacokinetics and ability to pass the blood–brain barrier (BBB) are important factors in the discovery of new agents for treating CNS disorders. There are now protocols for predicting the likelihood that a new compound will cross the BBB with appropriate metabolic stability, such as the multi-parameter optimization (MPO) score [97,98,99,100]. The MPO score estimation for CNS drugs, along with binding/functional affinity measurement could be useful tool to predict and design radiotracers for PET. Use of such algorithms will aid in the discovery of future A_2A_ antagonists for the treatment of CNS disorders, such as PD and AD, or for treatment of peripheral conditions in which CNS penetration would cause side effects. Nevertheless, it must be kept in mind that multiple physicochemical and biochemical factors influence the suitability of a given compound for use as a CNS drug or imaging agent [100].

Table 2 shows the MPO scores calculated using standard commercial software of common A_2A_ antagonists, including those in clinical trials and those for use as PET imaging agents [98,99]. This is a means of predicting which antagonists might be suitable for use in treatment of neurodegenerative diseases. For CNS drug candidates expected to be readily brain penetrant, the high clogBB (log([brain]:[blood]) value (>−0.2) and “no” substrate for the brain efflux pumps (e.g., P-gp), and a higher Intravenous CNS Profile Score [100] are preferable. A logP value <4 is one of many parameters predictive of effective oral dosing [100]. Thus, alkylxanthines caffeine and DMPX readily cross the BBB with clogBB values of −0.0323 and −0.180. Most of the compounds shown in Table 1 have the ability to cross the BBB to varying degrees, but an anti-tumor agent, inupadenant (clogBB −1.25), is not brain penetrant.

Furthermore, the application of in vivo imaging with PET ligands has become an important component in CNS drug development to determine target engagement and occupancy related to duration of drug action. Fortunately, there are numerous A_2A_ antagonist PET ligands developed over decades, resulting in [^18^F]MNI-444 ([^18^F] **12**) and [^11^C]preladenant ([^18^F] **11**), which demonstrated high specific binding signal [82] as clinical imaging tools. The interaction of A_2A_ antagonists and the D_2_ dopamine receptor was also studied in humans using istradefylline (40 mg QD) and PET imaging of dopamine receptors with [^11^C]raclopride [104].

## 4. Conclusions

This paper presents a brief overview of A_2A_ receptor antagonist clinical use in the context of drug discovery for AD therapy. Many new A_2A_ antagonists are being tested clinically, mostly for cancer, and some of these compounds are able to cross the BBB, which would enable their use in AD. Overexpression and functional upregulation of the A_2A_ receptor is associated with early changes leading to progressive neurodegeneration. Thus, PET ligands that are validated for A_2A_ receptor imaging in the human brain could be useful for the early clinical diagnosis of AD or other neurodegenerative conditions, which is an unmet medical need. Given the generality of the involvement of this receptor in brain injury and other pathology, the clinical development of A_2A_ receptor antagonists for treatment of these conditions promises to be on the verge of becoming a reality.

## Figures and Tables

**Figure 1 molecules-27-02680-f001:**
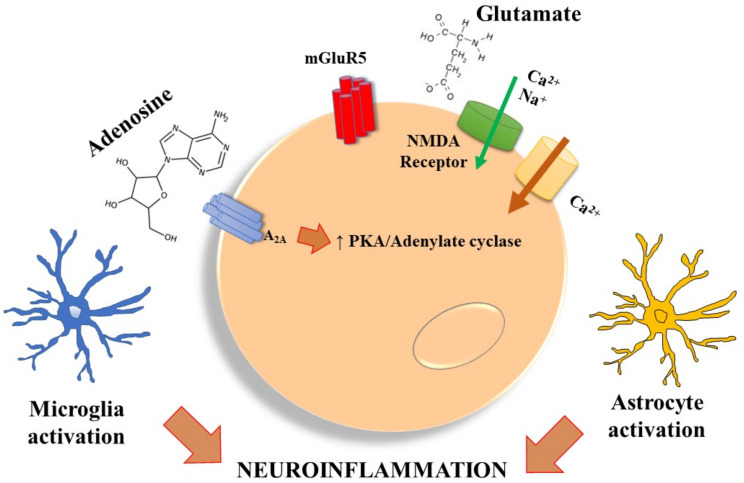
The picture represents the physiological effect of A_2A_ receptor activation on post-synaptic neurons, microglia, and astrocytes. In more detail, its post-synaptic stimulation leads to mGluR_5_ activation and NMDA phosphorylation thus increasing calcium influx, while in microglia and astrocytes it raises activation and proliferation with an increase in neuroinflammation.

**Figure 2 molecules-27-02680-f002:**
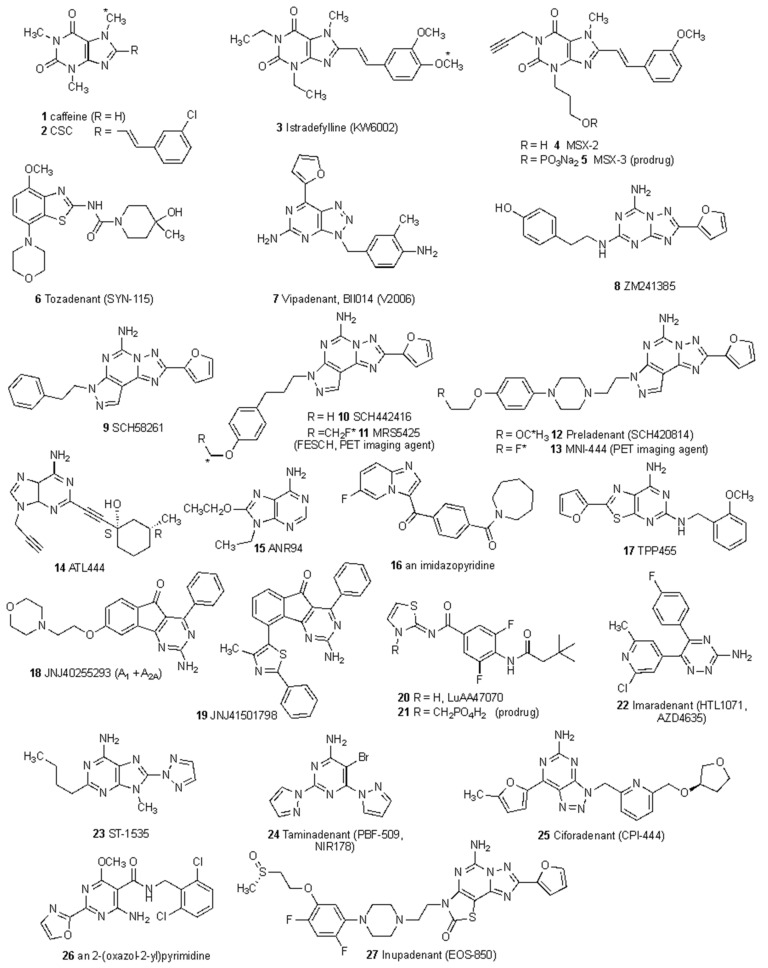
Structures of representative A_2A_ receptor antagonists **1**–**27**, including those listed in Table 1. Some of these pharmacological probes and clinical candidate compounds have been radiolabeled for PET. Other compounds not listed in Table 1 are described in the literature [67,68,69]. Additional dedicated PET ligands are shown in Figure 3. Compound **26** is one of a series of A_2A_ antagonists discovered by AdoRx Therapeutics [68]. * Indicates the location of a positron emitting isotope (^11^C or ^18^F) when used as a PET tracer [70].

**Figure 3 molecules-27-02680-f003:**
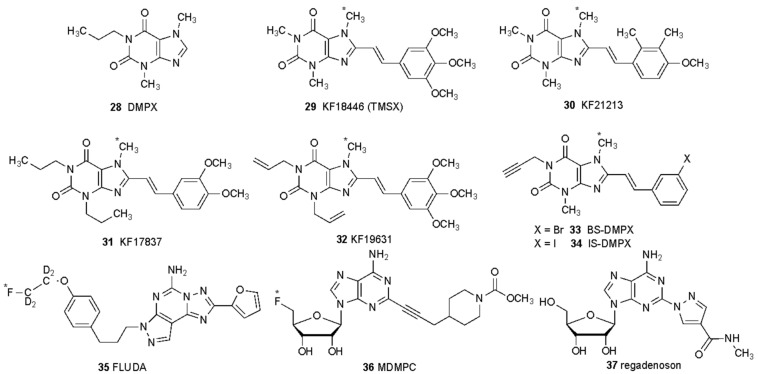
Structures of additional A_2A_ receptor ligands (antagonists **28**–**36**) that have been radiolabeled for PET imaging, as described in in Sun et al. [70] * Indicates the location of a positron emitting isotope (^11^C or ^18^F). DMPX **28** is shown for comparison as an early, slightly selective A_2A_ antagonist that is too low in affinity to be used a PET tracer. In addition, the only synthetic A_2A_ agonist approved for human use (regadenoson **37**, used in unlabeled form for myocardial perfusion imaging) is shown.

**Table 1 molecules-27-02680-t001:** A_2A_ receptor antagonists that are or have been in clinical trials for neurodegenerative diseases and/or for cancer immunotherapy. This list does not include compounds that display mixed selectivity for A_2A_ and A_2B_ receptors.

Compound (hA_2A_R Affinity, pKi)	Company or Sponsor (Country)	Condition (Reference, Dose)	Clinicaltrials.Gov Number (Phase)
Caffeine (**1**), (5.0, non-selective)	University Hospital, Lille (France)	AD [50] 200 mg BID	NCT04570085 (3)
Istradefylline (**3**, KW-6002), (7.44)	Kyowa Hakko Bio (Japan)	PD [71,72] 20 or 40 mg BID	NCT00250393 (2)
			NCT00955526, 6002–009 (3)
			NCT01968031 (3)
Tozadenant (**6**, SYN-115), (8.30)	Hoffmann-La Roche (The Switzerland); Biotie Therapies (Finland)	PD [73,74] 120 or 180 mg BID	NCT01283594 (2) NCT03051607 (3)
Vipadenant (**7**, BIIB014), (8.89)	Vernalis (UK); Biogen Idec, RedoxTherapies (Juno Therapeutics) (US)	PD [67,75] 30 or 100 mg QD	NCT00438607 (2) NCT00442780 (2)
Preladenant (**12**, MK-3814, SCH 420814), (8.96)	Merck (US)	PD, antipsychotic drug side effects, cancer [67] 5 mg BID	NCT01155479, PARADYSE (3) NCT00686699, P04628 (2) NCT0309916 (1)
Imaradenant (**22**, AZD4635, HTL1071), (8.77)	Sosei Heptares (Japan, UK), AstraZeneca (UK)	cancer [76] 75 mg QD	NCT03980821 (1) NCT03381274 (1/2) NCT02740985 (1) NCT04089553 (2)
Taminadenant (**24**, NIR178, PBF-509), (7.92)	Palobiofarma SL (Spain), Novartis (The Switzerland)	PD, cancer [77] 80–640 mg BID	NCT02111330 (1)
			NCT02403193, AdenONCO (1/2)
			NCT03207867 (3)
			NCT04895748 (1)
Ciforadenant (**25**, CPI-444, V81444), (8.45)	Bristol-Myers Squibb (US); Corvus Pharmaceuticals (US); Vernalis (UK)	cancer [78] 100 or 200 mg BID	NCT03454451 (1) NCT03337698 (1/2) NCT03549000 (1)
Inupadenant (**27**, EOS100850, EOS-850)	iTeos (Belgium, US)	cancer [79] 20 mg QD–160 mg BID	NCT03873883 (1) NCT05117177 (1)
TT-10 (structure not disclosed)	Tarus Therapeutics (US)	cancer [80] 10–200 mg BID	NCT04969315 (1/2)
CS3005 (structure not disclosed)	CStone (China)	Cancer [81]	NCT04233060 (1)

Abbreviations: AD, Alzheimer’s disease; PD, Parkinson’s disease; PD-1, programmed cell death 1.

**Table 2 molecules-27-02680-t002:** Calculated MPO parameters of representative A_2A_ antagonists and one agonist **37** as shown in Figure 1 and Figure 2, including compounds that are for use as PET tracers. The probabilistic scores indicative of CNS penetration were calculated for each small molecule with the StarDrop software package (Optibrium Inc., Cambridge, UK) [98,99]. Six physicochemical parameters that affect BBB passage are included in this calculation: molecular weight, most basic center’s p*K*_a_, calculated logP (clogP), calculated logD at pH 7.4 (clogD), topological polar surface area (tPSA), and hydrogen bond donor number. The BBB log([brain]:[blood]) (clogBB) parameter is a single indicator of probability of crossing the BBB at pharmacologically significant levels.

Number	Compound [Reference]	Intravenous CNS Scoring Profile Score	Predicted BBB Log([Brain]:[Blood])	P-gp Category	logP
**1**	caffeine [12]	0.355	−0.0323	no	0.0231
**2**	CSC [10]	0.238	−0.539	no	2.24
**3**	istradefylline [73]	0.129	−0.999	yes	2.16
**4**	MSX-2 [10]	0.119	−1.25	no	1.12
**6**	tozadenant [74,92]	0.0786	−0.891	yes	2.47
**7**	vipadenant [75]	0.103	−0.388	yes	2.12
**8**	ZM241,385 [70]	0.101	−1.10	yes	2.13
**9**	SCH58241 [70]	0.0783	−0.644	yes	2.60
**10**	SCH442416 [94]	0.0926	−0.779	yes	2.78
**11**	MRS5425 [70]	0.0747	−0.773	yes	3.08
**12**	preladenant [95]	0.105	−0.592	yes	2.41
**13**	MNI-444 [70]	0.0964	−0.495	yes	2.73
**14**	ATL444 [101]	0.163	−1.17	no	1.16
**15**	ANR94 [101]	0.147	−0.953	no	0.958
**16**	imidazopyridine [101]	0.113	0.0684	no	3.47
**17**	TPP455 [11]	0.0753	−0.876	no	3.14
**18**	JNJ40255293 [10]	0.121	−0.219	yes	2.72
**19**	JNJ41501798 [102]	0.0200	−0.600	no	5.14
**20**	LuAA47070 [101]	0.122	−0.672	no	2.87
**21**	imaradenant [76]	0.0833	−0.508	no	3.04
**23**	ST-1535 [10]	0.0767	−1.07	yes	1.46
**24**	taminadenant [77]	0.153	−1.01	no	1.02
**25**	ciforadenenant [78]	0.0507	−0.965	yes	2.21
**26**	2-(oxazol-2-yl)pyrimidine [68]	0.0723	−0.502	no	2.64
**27**	inupadenant [79]	0.0994	−1.25	yes	2.66
**28**	DMPX [70]	0.308	−0.180	no	0.629
**29**	KF18446 [70]	0.227	−0.947	no	1.34
**30**	KF21213 [70]	0.183	−0.826	no	2.34
**31**	KF17837 [70]	0.178	−1.02	yes	2.55
**32**	KF19631 [70]	0.183	−0.525	yes	1.74
**33**	BS-DMPX [70]	0.192	−0.603	no	2.21
**34**	IS-DMPX [70]	0.185	−0.616	no	2.26
**35**	FLUDA [103]	0.0674	−0.774	yes	3.08
**36**	MDMPC [70]	0.0934	−1.43	yes	0.678
**37**	regadenoson [101]	0.0551	−1.30	yes	−1.35

## Data Availability

Not applicable.

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
