# Peer review of "Pathophysiological Role and Medicinal Chemistry of A_2A_ Adenosine Receptor Antagonists in Alzheimer’s Disease"

_molecules, 2022, doi:10.3390/molecules27092680_

Round 1
Reviewer 1 Report
well organised and well written paper
Author Response
We thank the reviewer for his/her kind comments
Reviewer 2 Report
Merighi et al. described pathological roles of A2a adrenosine receptors in neurodegenerative diseases, including Alzheimer’s disease. Authors also summarized lists of A2a adrenosine receptor antagonists that are currently in clinical trials. Overall, this review article is informative and provides an optimistic point of view of A2a adrenosine receptor antagonists as a potential pharmaceutical target for neurodegenerative diseases and cancer.
Lists provided below are a few suggestions for authors to supplement the context.
- On page 2, line 94-96: “However, due to the failure of numerous clinical trials with drugs directed towards Abeta protein, it is questionable whether beta-amyloid is certainly an effective target.”. Authors should discuss on several lines of evidence suggested that tau neurofibrillary tangles are strongly correlated to local neurodegeneration and cognitive impairment in AD. Also, they should provide the relation of tau pathology on A2a adrenosine receptors in AD and other tauopathies.
- In Figure 1, authors should revise the figure and figure legend. Should it be a comparison of A2a adrenosine receptor functions/ expressions between physiological conditions versus pathological conditions, i.e. AD pathology?
- Table 2 should include their references.
- After conclusion, authors should provide a brief discussion on the future perspective of A2a adrenosine receptor antagonists for clinical diagnosis or early biomarkers for neurodegenerative diseases or cancers.
Author Response
Please find in attach the reviewer's response.

Reviewer 3 Report
Ref: molecules-1665102
Title: Pathophysiological role and medicinal chemistry of A2A adenosine receptor antagonists in Alzheimer’s disease
Recommendation: Accept for publication
Very well written review article. The topic of the paper is not novel, but the Authors refer to the latest reports in the field. For this reason, the review will certainly find many readers. Therefore, I recommend this paper to be publish as it is.
I comment - please be more specific and add more informations to the Abstract section. It is worth remembering that Abstract is the part that decides whether other scientists will cite the manuscript or not.
Author Response

(The authors gave the same response as above.)
